# Inhibition of Soluble Epoxide Hydrolase Does Not Promote or Aggravate Pulmonary Hypertension in Rats

**DOI:** 10.3390/cells12040665

**Published:** 2023-02-20

**Authors:** Matthieu Leuillier, Valentin Platel, Ly Tu, Guillaume Feugray, Raphaël Thuillet, Déborah Groussard, Hind Messaoudi, Mina Ottaviani, Mustapha Chelgham, Lionel Nicol, Paul Mulder, Marc Humbert, Vincent Richard, Christophe Morisseau, Valéry Brunel, Thomas Duflot, Christophe Guignabert, Jérémy Bellien

**Affiliations:** 1INSERM EnVI UMR 1096, Health Campus, University of Rouen Normandie, F-76000 Rouen, France; 2INSERM UMR_S 999, Hôpital Marie Lannelongue, F-92350 Le Plessis-Robinson, France; 3Faculté de Médecine, Université Paris-Saclay, F-94276 Le Kremlin-Bicêtre, France; 4Department of General Biochemistry, CHU Rouen, F-76000 Rouen, France; 5Department of Pharmacology, CHU Rouen, F-76000 Rouen, France; 6Department of Entomology and Nematology, UCD Comprehensive Cancer Center, University of California, Davis, CA 95616, USA

**Keywords:** soluble epoxide hydrolase, epoxyeicosatrienoic acids, pulmonary arterial hypertension, right ventricular dysfunction

## Abstract

Inhibitors of soluble epoxide hydrolase (sEH), which catalyzes the hydrolysis of various natural epoxides to their corresponding diols, present an opportunity for developing oral drugs for a range of human cardiovascular and inflammatory diseases, including, among others, diabetes and neuropathic pain. However, some evidence suggests that their administration may precipitate the development of pulmonary hypertension (PH). We thus evaluated the impact of chronic oral administration of the sEH inhibitor TPPU (N-[1-(1-Oxopropyl)-4-piperidinyl]-N′-[4-(trifluoromethoxy)phenyl]-urea) on hemodynamics, pulmonary vascular reactivity, and remodeling, as well as on right ventricular (RV) dimension and function at baseline and in the Sugen (SU5416) + hypoxia (SuHx) rat model of severe PH. Treatment with TPPU started 5 weeks after SU5416 injection for 3 weeks. No differences regarding the increase in pulmonary vascular resistance, remodeling, and inflammation, nor the abolishment of phenylephrine-induced pulmonary artery constriction, were noted in SuHx rats. In addition, TPPU did not modify the development of RV dysfunction, hypertrophy, and fibrosis in SuHx rats. Similarly, none of these parameters were affected by TPPU in normoxic rats. Complementary in vitro data demonstrated that TPPU reduced the proliferation of cultured human pulmonary artery-smooth muscle cells (PA-SMCs). This study demonstrates that inhibition of sEH does not induce nor aggravate the development of PH and RV dysfunction in SuHx rats. In contrast, a potential beneficial effect against pulmonary artery remodeling in humans is suggested.

## 1. Introduction

Soluble epoxide hydrolase (sEH) is a ubiquitous enzyme metabolizing biologically active epoxy fatty acids (EpFAs), synthetized from n-3 and n-6 fatty acid precursors by CYP450s, to their corresponding diols. In particular, sEH converts the vasodilator and anti-inflammatory arachidonic acid derivatives epoxyeicosatrienoic acids (EETs) to dihydroxyeicosatrienoic acids (DHETs), which are biologically less active [1,2]. Based on an increasing amount of experimental evidence showing their beneficial effects in animal disease models [1,2], sEH inhibitors have recently entered the first phases of clinical development for the treatment of various diseases, including hypertension, diabetes, stroke, dyslipidemia, pain, immunological disorders, neurological diseases, eye diseases, and other indications [3,4]. However, in contrast to the systemic circulation, EETs were shown to exert deleterious effects on the pulmonary vasculature. In particular, EETs were shown to potentiate the pulmonary contractile response by recruiting transient receptor potential C6 channels to caveolae and exogenous EET administration was shown to enhance right ventricular (RV) systolic pressure in vivo [5,6]. In addition, EETs contribute to hypoxic pulmonary hypertension in mice and selective epoxygenase inhibition significantly reduces acute hypoxic pulmonary vasoconstriction and chronic hypoxia-induced pulmonary vascular remodeling [7]. Therefore, it is suspected that the development of pulmonary hypertension (PH) may represent a potential serious side effect of sEH inhibition mediated by the increase in EETs.

PH is a fatal and incurable disease defined by an elevated resting mean pulmonary artery pressure (mPAP) > 20 mmHg, measured by right heart catheterization [8,9]. This increase in mPAP can result from pre-capillary (arterial) or post-capillary (cardiac/venous) pathophysiological mechanisms. The current clinical classification divides PH into five groups based on pathophysiology and clinical features in order to optimize therapeutic approaches, predict patient outcomes, and facilitate research strategies [8,9]. Right heart failure is a major cause of morbidity and mortality in PH.

Nevertheless, sEH inhibition was shown to not affect muscularization of the precapillary pulmonary arterioles in response to chronic hypoxia in mice [10] and to delay, but not prevent, pulmonary arterial remodelling and PH development induced by monocrotaline in rats [9]. In the latter study, experimental evidence suggests that these beneficial effects could be related to an indirect effect of EETs on the acute inflammation known to contribute to the development of PH in this specific animal model [11]. For all of these reasons, this study was designed to thoroughly assess the impact of sEH inhibition in the Sugen (SU5416) + hypoxia (SuHx) rat model of severe PH that more closely recapitulates the human disease, particularly in terms of obliterative arteriopathy; progression of pulmonary vascular disease; and progressive, severe RV dysfunction, ultimately resulting in RV failure [12,13]. Complementary experiments were conducted to study the in vitro and in vivo effects of sEH inhibition on the proliferation of primary human pulmonary artery smooth muscle cells (hPA-SMCs) and on pulmonary hemodynamic parameters and vascular remodelling in Sprague–Dawley rats, respectively.

## 2. Materials and Methods

### 2.1. Animals and Treatment

All animal care and procedures were approved by French Animal Experimentation Ethics Committees and performed in accordance with the guidelines from the French National Research Council for the Care and Use of Laboratory Animals (permit numbers: Apafis #24107 and #11920). Experiments were performed in 10-week-old male wild-type Sprague–Dawley rats (Janvier Labs) rats. Male rats were used to minimize hormonal effects (e.g., estrogen). Rats received a single subcutaneous administration of the VEGF receptor antagonist SU5416 (20 mg/kg) and were then exposed to hypoxia (10% FiO_2_) for 3 weeks [12,13]. Then, these rats returned to normoxia (21% FiO_2_) for 5 additional weeks before evaluation. Inhibition of sEH was achieved in SuHx rats using the oral administration of 1-trifluoromethoxyphenyl-3-(1-propionylpiperidin-4-yl) urea (TPPU; 5 mg/L in drinking water/PEG400, 99:1, *v*/*v*). TPPU treatment was started 5 weeks post-SU5416 injection and went on for 3 weeks. TPPU was provided by C.M. for the study. High in vivo exposure and sEH inhibitory potency, in particular at the cardiac level, of TPPU have already been shown at the dose of 5 mg/L of drinking water in adult male Sprague–Dawley rats [14]. Normoxic rats receiving vehicle or TPPU served as controls.

### 2.2. Cardiovascular Parameters Investigated

Pulsed-wave Doppler during transthoracic echocardiography was used in anesthetized rats to evaluate pulmonary artery acceleration time (AT) to RV ejection time (ET) ratio, using Vivid E9 (GE Healthcare, Velizy-Villacoublay, France).

Briefly, end-diastolic and end-systolic RV volumes were assessed in methohexital-anesthetized animals using a Bruker Biospec 4.7 Tesla MRI, and Cinema MRI was obtained using the acquisition sequence Intragate (Bruker France, Wissembourg, France). Briefly, the animals were positioned prone on an actively decoupled tray and warming pad with hot water circulation used to maintain its physiological temperature. For the assessment of RV volumes, whole heart coverage with contiguous short-axis cine imaging (10–12 slices; 12 phases) was carried out followed by the acquisition of standard two-chamber views. Data analysis was performed with an operator-interactive threshold technique by one trained observer using FARM-CAAS 2.0 software (Pie Medical, Maastricht, The Netherlands). RV volumes were determined from end-diastolic and end-systolic images by multiplication of the RV compartment area and slice thickness (1.5 mm). Total volumes were calculated as the sum of all slice volumes. Ejection fraction (EF) was calculated with the end-diastolic (EDV) and end-systolic (ESV) volumes (EF = SV/EDV).

RV hemodynamic measurements were performed in anesthetized rats (2.0% in room air), using a polyvinyl catheter introduced into the right external jugular vein; advanced in the right ventricle; and, further, in the pulmonary artery, allowing the measurement of RV systolic pressure and mean pulmonary arterial pressure (mPAP). In addition, the cardiac output (CO) was measured by thermodilution, and total pulmonary resistance was calculated using the ratio of mPAP/CO. After measurement of hemodynamic parameters, the rat was euthanized (by exsanguination under isoflurane) and the thorax was opened. The left lung was immediately removed and frozen. The right lung was fixed in the distended state with formalin buffer. RV hypertrophy was assessed by the Fulton index and the percentages of muscularized vessels were calculated. Immunohistochemistry and immuno-cytofluorescent staining with antibodies against α-smooth muscle cell actin (α-SMA) and proliferating cell nuclear antigen (PCNA) (Dako, Les Ulis, France), smooth muscle (SM)22, and CD68 (Santa Cruz Biotechnology, Le Perray-en-Yvelines, France) were performed as previously described [13,15,16]. In addition, total collagen content of the right ventricle was assessed by Picrosirius red staining.

For the evaluation of pulmonary artery reactivity, rats were anesthetized (10 mg/kg xylazine, 90 mg/kg ketamine) and the unilobar left lung was removed and immediately placed in cold, oxygenated Krebs buffer. A segment of second-order intralobar pulmonary artery, 2.5 to 3.7 mm long and ~450 µm in diameter, was carefully dissected and mounted in a small vessel myograph for isometric tension recording. The time-dependent contractile response to 10^−5^ M phenylephrine was assessed.

### 2.3. Cardiovascular Parameters Investigated

Blood samples were drawn on EDTA tubes, immediately centrifuged (4500× *g*, 5 min, 4 °C), and the plasma was frozen at −80 °C until analysis. Plasma quantification of EET and DHET regioisomers as well as the levels of EpFAs synthesized by CYP450 from the n-3 docosahexaenoic acid (epoxydocosapentaenoic acids (EpDPAs)) and eicosapentaenoic acid (eicosatetraenoic acids (EpETE)) and from the n-6 linoleic acid (epoxyoctadecenoic acids (EpOMEs)), as well as their respective diols produced by sEH, dihydroxydocosapentaenoic acids (DiHDPA), dihydroxytetraenoic acids (DiHETE), and dihydroxyoctadecenoic acids (DiHOMEs), were quantified using a previously developed liquid chromatography coupled with tandem mass spectrometry method [17].

#### 2.3.1. Chemicals and Reagents

Ethyl acetate (EA), methanol (MeOH), and water of HPLC grade were purchased from Carlo Erba (Fontenay aux-Roses, France). Acetic acid was purchased from Carlo Erba (Fontenay-aux-Roses, France). Solution stocks and deuterated internal standards (IS) (14,15-EETd11, 14,15-DHETd11, and 15-HETEd8) were purchased from Bertin technologies (Montigny-le-Bretonneux, France). Protein LoBind^®^ (1.5 mL) tubes were purchased from Eppendorf (Hamburg, Germany). Chromatographic Kinetex^®^ C18 column (50 mm L × 3 mm inner diameter, 2.6 µm) was purchased from Phenomenex (Le Pecq, France). Oasis HLB-SPE-columns (3 mL, 60 mg, 30 µm particles) were purchased from Waters (Guyancourt, France).

#### 2.3.2. Sample Preparation

Analytes in their free form were extracted from plasma using a deproteinized step prior to a solid phase extraction (SPE). First, 10 µL of IS solution in MeOH (30 ng/mL of 14,15-DHETd11, 14,15-EETd11, and 15-HETEd8) was added to 500 µL of plasma. Then, the sample was deproteinized by adding 990 µL of MeOH and vortexed. After a centrifugation step (5 min, 20,000× *g*), the supernatant was loaded into a preconditioned Oasis HLB-SPE-column. The column was washed with 6 mL MeOH/water (5/95, *v*/*v*) and the cartridge was dried for 20 min. Oxylipins were eluted by gravity into glass tubes with 0.5 mL MeOH and 1.5 mL EA. Then, eluted compounds in MeOH/EA were evaporated under nitrogen for 20 min at room temperature. The residue was reconstituted with 50 µL of MeOH, vortexed, and transferred into an autosampler vial, and 10 µL was injected into the UHPLC–MS/MS system.

#### 2.3.3. LC-MS/MS Conditions

Oxylipin assays were performed on a UHPLC–MS/MS system consisting of the following Shimadzu^®^ modules (Shimadzu Corporation, Marne-la-Vallée, France): a binary pump consisting of coupling two isocratic pumps Nexera LC30AD, an automated sampler SIL-30AC, a column oven CTO-20AC, and a triple-quadrupole mass spectrometer LCMS-8060 operating in the negative ion mode. Chromatographic separation was achieved on a Kinetex^®^ C18 column maintained at 50 °C and gradient-elution chromatography using the following mobile phases: water with 0.01% acetic acid (A) and methanol (B) at a flow rate of 0.600 mL/min. Gradient was as follows: 0.0–0.5 min, 10% (B); 0.5–2.0 min, 10 to 70% (B); 2.0–5.0 min, 70 to 75% (B); 5.0–5.1 min, 75 to 98% (B); 5.1–6.9 min, 98% (B); 6.9–7.0 min, 98% to 10% (B); and 7.0–8.0 min, 10% (B). The source interface parameters and common settings were as follows: interface voltage: −3 kV; nebulizing gas flow: 3 L/min; heating gas flow: 10 L/min; drying gas flow: 10 L/min; interface temperature: 400 °C; DL (desolvation line) temperature: 250 °C; heat block temperature: 500 °C; and collision gas pressure: 300 kPa. Detection and quantification were performed by scheduled-MRM (multiple reaction monitoring) using a pause time of 3 ms and individual dwell times to achieve sufficient points per peak. Isobaric compounds of 14,15-DHET, 14,15-EET, and 15-HETE were used as IS.

### 2.4. Proliferation of Human PA-SMCs

This study was approved by the local ethics committee (CPP Est-III: N°ID RCB: 2018-A01252-53, N° CPP: 18.06.06) and all patients gave informed consent before the study. Lung specimens were obtained during lobectomy or pneumonectomy for localized lung cancer. The lung specimens were collected at a distance from the tumor foci. Human PA-SMCs were isolated and cultured in the absence and presence of 5% foetal bovine serum (FBS), as previously described [13]. PA-SMC proliferation was measured by BrdU incorporation and by cell counting. BrdU staining was measured by the DELFIA^®^ Cell proliferation kit (PerkinElmer, Courtaboeuf, France) and Time-resolved fluorometer EnVisionTM Multilabel Reader (PerkinElmer) after 24 h of culture without and with TPPU at 1 and 10 µM.

### 2.5. Statistical Analyses

Data are presented as mean ± SEM. Group effects were determined by one-way ANOVA or repeated measures ANOVA with Tukey’s post hoc test. A value of *p* < 0.05 was considered statistically significant. Statistical analyses were conducted using GraphPad Prism software 8.0.2.263.

## 3. Results

### 3.1. sEH Inhibition Does Not Alter Hemodynamic and Pulmonary Hemodynamic Parameters in Healthy Sprague–Dawley Rats

First, the effects of chronic treatment of TPPU on hemodynamic and pulmonary hemodynamic parameters were examined in healthy Sprague–Dawley rats. No difference was found in mPAP, TPVR, AT/ET, and CO in rats treated with TPPU for 3 weeks when compared with vehicle-treated rats (Figure 1A–D).

Consistent with these findings, no difference was observed in the Fulton index, nor in accumulation of collagen (stained with picosirius red) in the RV, nor in the percentage of muscularized arteries (Figure 2A–C).

The numbers of CD68 and PCNA positive cells were also similar (Figure 3A,B).

In addition, RV end-diastolic diameter and RV ejection fraction were not affected by TPPU (Figure 4A,B).

Next, we tested whether chronic treatment of TPPU may alter the pulmonary vascular reactivity to phenylephrine and found that TPPU did not modify phenylephrine-induced pulmonary artery constriction in normoxic rats (Figure 4C).

Taken together, these findings support the notion that sEH inhibition does not alter hemodynamic and pulmonary hemodynamic parameters in healthy Sprague–Dawley rats and did not affect RV structure and function.

### 3.2. sEH Inhibition Does Not Alter the Progression of PH in the SUGEN + Hypoxia (SuHx) Rat Model

Second, the effect of chronic treatment of TPPU against the progression of PH induced in rats was examined using the combination of a vascular endothelial growth factor receptor antagonist, Sugen (SU5416), and chronic hypoxia.

Five weeks after SU5416 + hypoxia treatment, non-invasive and invasive hemodynamics using right heart catheterization confirmed the presence of PH in vehicle-treated rats, as reflected by the increase in mPAP and TPVR and the decrease in pulmonary artery AT/ET and CO compared with control rats (Figure 1A–D). Consistent with these results, vehicle-treated SuHx rats also exhibited an increase in the Fulton index, in RV picrosirius staining, and in the percentages of muscularized distal pulmonary arteries together with higher numbers of CD68 and PCNA positive cells (Figure 2A–C and Figure 3A,B). Furthermore, RV dilatation and dysfunction was shown in vehicle-treated SuHx rats by the increase in RV end-diastolic diameter and decrease in RV ejection fraction (Figure 4A,B). Moreover, there was an abolishment of phenylephrine-induced pulmonary artery constriction in vehicle-treated SuHx rats (Figure 4C).

No differences in the values of mPAP, TPVR, pulmonary artery AT/ET, and CO were noted between vehicle-treated versus TPPU-treated SuHx rats (Figure 1A–D). Consistent with these results, TPPU did not affect the development of RV hypertrophy, fibrosis, dilatation, and dysfunction induced by SuHx, as shown by the similar increase in the Fulton index, in RV picrosirius staining, and in end-diastolic volume and by the similar decrease in RV ejection fraction in non-treated and treated rats (Figure 2 and Figure 3). Finally, consistent with the data obtained in control rats, TPPU did not prevent the abolishment of the contractile response in SuHx rats (Figure 4C).

Taken together, these findings support the notion that sEH inhibition neither promoted nor aggravated PH in rats induced by SuHx. This was observed while TPPU treatment only abolished the conversion of some EpFAs, i.e., 16(17)-EpDPA, 17(18)-EpETE, and 12(13)-EpOME, to their corresponding diols in control rats (Figure 5A–D). This effect was less marked in SuHx rats and conversions of 8,9-EET, 14,15-EET, 10,11-EpDPA, and 19,20-EpDPA were unexpectedly increased besides TPPU treatment. It should be noted that the levels of EpFA precursors, in particular arachidonic acid, were reduced by TPPU and SuHx treatment groups (Figure 5E).

### 3.3. sEH Inhibition Attenuates Proliferation of Cultured Human PA-SMCs

PA-SMC proliferation contributes to the progression of PH. The effect of TPPU treatment was thus examined on the proliferative potential of cultured human PA-SMCs. BrdU incorporation assay showed that TPPU (1 and 10 µM) had no significant effect on PA-SMC proliferation in the absence of serum (0% fetal bovine serum), whereas it partially prevented the increase in PA-SMC proliferation induced by serum 5% FBS (Figure 6).

## 4. Discussion

To the best of our knowledge, this is the first study assessing the effect of sEH inhibition by TPPU in the Sugen (SU5416) + hypoxia (SuHx) rat model of severe PH. The data indicate that TPPU neither promoted nor aggravated the development of PAH in rats and did not alter the pulmonary vascular reactivity to phenylephrine in both control and SuHx rats.

Inhibitors of sEH are currently under clinical development for the treatment of various diseases. This study was designed to assess whether the use of this new pharmacological class can produce a risk of developing PH in particular owing to the vasoconstrictor effect of EETs at the level of the pulmonary vasculature [5,6,7]. This has to be balanced with the potent anti-inflammatory action of EETs that has been suggested, in contrast, to contribute to delaying PH development in the monocrotaline rat model [11]. Because monocrotaline-induced inflammation is the main driver of PH development, the determination of the impact of sEH inhibition in the SuHx model, which mimics severe human PAH with occlusive neointimal lesions, was of critical importance. For this objective, the previously validated potent sEH inhibitor TPPU and state-of-the-art methods including RV catheterization, MRI, and histological analyses were used.

As expected, SuHx rats displayed all of the hallmarks of PAH, including an increase in pulmonary artery pressure, pulmonary artery remodeling with increased muscularization, cell proliferation and inflammation, as well as depressed phenylephrine sensitivity compared with vehicle-injected rats. In addition, RV hypertrophy, fibrosis, and dysfunction were evident in SuHx rats. In this context, TPPU was without an effect in vehicle-injected rats and, in particular, did not promote an increase in pulmonary arterial pressure, as previously observed using acute administration of exogenous EETs [6]. Moreover, TPPU neither prevented nor aggravated the development of PH in SuHx rats, as no change in the increase in mPAP, remodeling, and reactivity to phenylephrine was observed and RV abnormalities were also unaffected. Thus, in contrast to the previously demonstrated involvement of the CYP450/EETs/sEH axis in the acute response to hypoxia [7,10], the prolonged inhibition of sEH did not exacerbate PAH development. These results extend previous data obtained in mice under normoxia or submitted to chronic hypoxia, showing the absence of increased pulmonary artery and RV remodeling when inhibitors of sEH were chronically administrated [18]. In addition, a phase 1 clinical trial showed that 14 days of repeat dosing of the sEH inhibitor GSK2256294 did not affect pulmonary artery pressure in healthy subjects, estimated using transthoracic echocardiography [3].

The results of the present study were observed while the conversion of EETs appeared to not be affected in control rats and even decreased in SuHx rats receiving TPPU. An increased sEH expression is unlikely to contribute to this unexpected finding because chronic hypoxia, in contrast, was shown to reduce its expression [18]. However, it may be due to a compensatory increase in microsomal EH (mEH) activity, whose involvement in the metabolism of some EpFA may increase during sEH inhibition, in particular when the arachidonic acid level is low, as in the present work, together with modifications in metabolic enzyme gene expression, notably CYP450 [19,20,21]. The complex impact of sEH inhibition on protective n-3-derived EpFAs and on pro-inflammatory DiHOME formation may also explain the neutral effect of sEH inhibition on PAH development. For instance, it was shown that chronic administration of n-3 epoxides prevented PAH development in SuHx mice, through in particular suppression of advanced pulmonary vascular remodeling, while 14,15-EET did not [22]. Finally, the in vitro experiments showed that TPPU efficiently reduces the proliferation of human PA-SMC, which could be translated in the long term by a protective effect against the development of PAH in patients treated with sEH inhibitors. In addition, sEH inhibition was shown to preserve cardiac contractility, in particular during ischemia-reperfusion injury, through EET-mediated modulation of L-type calcium and ATP-sensitive potassium channels [23], which may also be protective in the setting of PAH and RV dysfunction.

Together, the results show that chronic sEH inhibition did not promote the development of PH nor of its consequences on the right ventricle in rats. These data provide new convincing evidence on the safety of sEH inhibitors to be used for the treatment of human chronic diseases.

## Figures and Tables

**Figure 1 cells-12-00665-f001:**
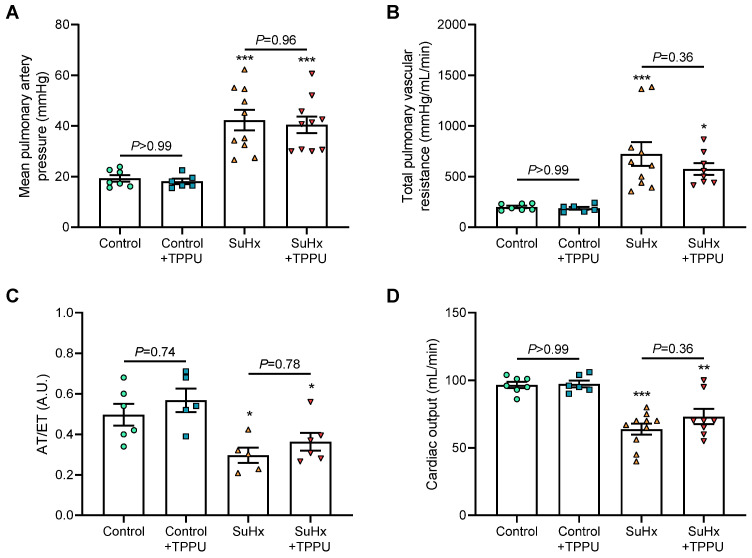
Values of mean pulmonary arterial pressure (**A**), total pulmonary vascular resistance (**B**), pulmonary artery acceleration time to right ventricular ejection time ratio (AT/ET; (**C**)), and cardiac output (**D**) obtained in normoxic control rats and in rats receiving Sugen 5416 associated with 3-week hypoxia (SuHx) treated with vehicle or with the soluble epoxide hydrolase inhibitor TPPU. AT/ET was obtained in a subset of animals from each group. Two animals died in the SuHx + TPPU group during hemodynamical measurements before cardiac output and pulmonary vascular resistance could be obtained. * *p* < 0.05, ** *p* < 0.01, and *** *p* < 0.001 vs. corresponding control rats.

**Figure 2 cells-12-00665-f002:**
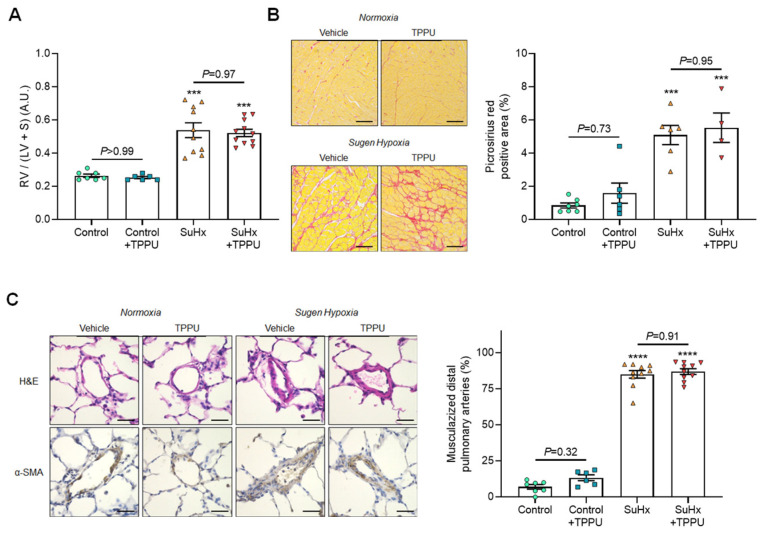
Values of the Fulton index (right ventricular (RV) weight divided by left ventricular + septum weight (LV+S)) (**A**); representative images and quantification of Picrosirius red staining of tissue section of right ventricles (**B**); representative images of haematoxylin and eosin-, α-smooth muscle actin, and quantification of the percentage of muscularized distal pulmonary arteries (**C**), obtained in normoxic control rats and in rats receiving Sugen 5416 associated with 3-week hypoxia (SuHx) treated with vehicle or with the soluble epoxide hydrolase inhibitor TPPU. One SuHx + TPPU rat died before hemodynamical measurements, presented in Figure 1. Sirius red staining was obtained in a subset of animals from each group. Scale bar: 20 µm. *** *p* < 0.001, **** *p* < 0.0001 vs. corresponding control rats.

**Figure 3 cells-12-00665-f003:**
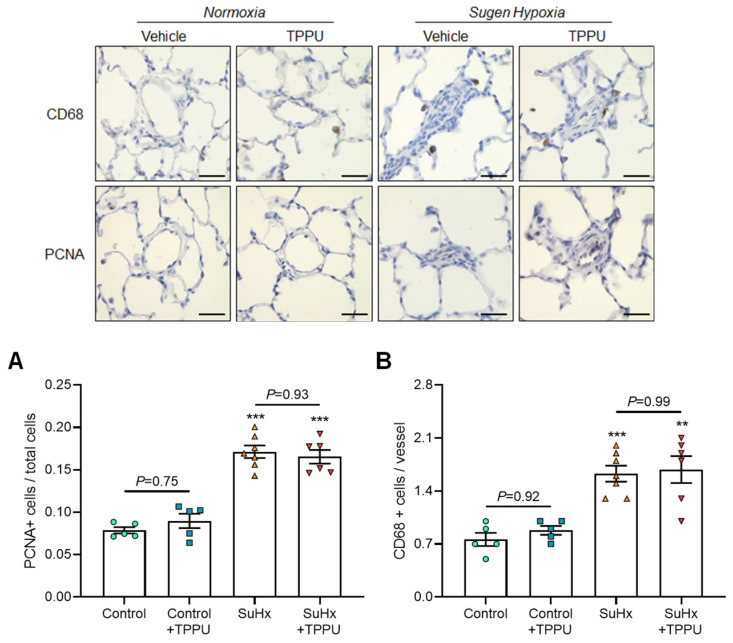
Representative images of proliferating cell nuclear antigen (PCNA) and CD68-stained sections of distal pulmonary arteries and quantification of the number of PCNA (**A**) and CD68 (**B**) positive cells per vessel obtained in normoxic control rats and in rats receiving Sugen 5416 associated with 3-week hypoxia (SuHx) treated with vehicle or with the soluble epoxide hydrolase inhibitor TPPU. Data were obtained in a subset of animals from each group. Scale bar: 20 µm. ** *p* < 0.01, *** *p* < 0.001 vs. corresponding control rats.

**Figure 4 cells-12-00665-f004:**
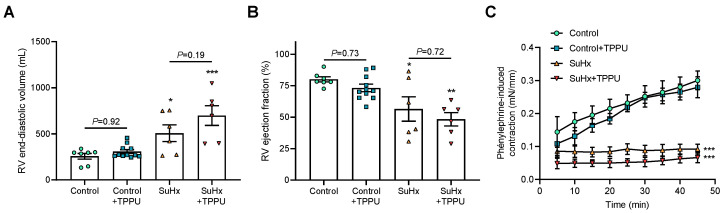
Values of RV end-diastolic volume (**A**), RV ejection fraction (**B**), and phenylephrine-induced pulmonary artery constriction (**C**) obtained in normoxic control rats and in rats receiving Sugen 5416 associated with 3-week hypoxia (SuHx) treated with vehicle or with the soluble epoxide hydrolase inhibitor TPPU. Data were obtained from different series of animals than those presented in Figure 1, Figure 2 and Figure 3. * *p* < 0.05, ** *p* < 0.01, and *** *p* < 0.001 vs. corresponding control rats.

**Figure 5 cells-12-00665-f005:**
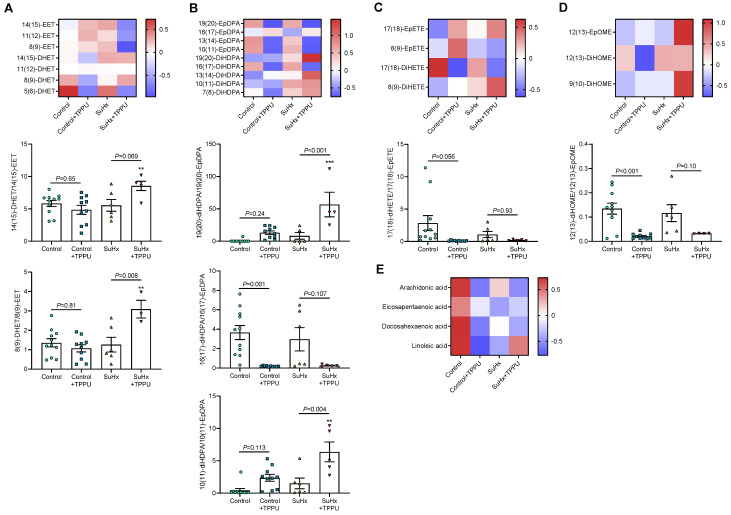
Z-score heat maps of plasma levels and diol-to-epoxide ratios of epoxyeicosatrienoic acid (EET) and dihydroxyeicosatrienoic acid (DHET) regioisomers (**A**); epoxydocosapentaenoic acid (EpDPA) and dihydroxydocosapentaenoic acid (DiHDPA) regioisomers (**B**); eicosatetraenoic acid (EpETE) and dihydroxytetraenoic acid (DiHETE) regioisomers (**C**); epoxyoctadecenoic acid (EpOME) and dihydroxyoctadecenoic acid (diHOME) regioisomers (**D**); and plasma levels of their respective precursors, i.e., arachidonic acid, docosahexaenoic acid, eicosapentaenoic acid, and linoleic acid (**E**), obtained in normoxic control rats and in rats receiving Sugen 5416 associated with 3-week hypoxia (SuHx) treated with vehicle or with the soluble epoxide hydrolase inhibitor TPPU. Data were obtained from the same series of animals as presented in Figure 4. Only statistically significant diol-to-epoxide ratios are shown (*p* < 0.05, ANOVA). ** *p* < 0.01 and *** *p* < 0.001 vs. corresponding control rats.

**Figure 6 cells-12-00665-f006:**
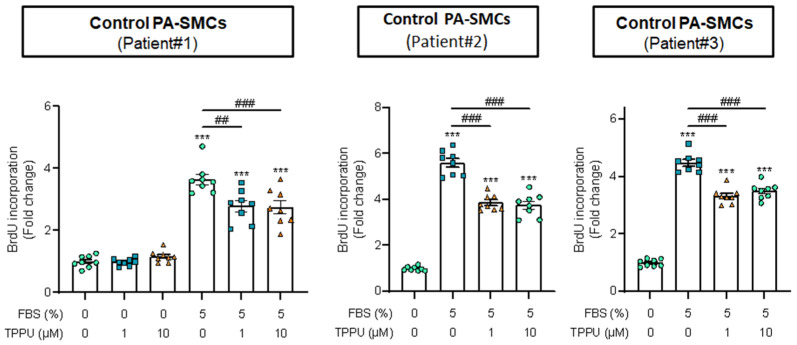
Impact of the soluble epoxide hydrolase inhibitor TPPU on the proliferation of human pulmonary artery smooth muscle cell (PASMC) assessed using 5-bromo-2-deoxyuridine (BrdU) incorporation. Technical replicates with PASMCs derived from three different patients. *** *p* < 0.001 vs. 0% FBS; ## *p* < 0.01 and ### *p* < 0.001 vs. 5% FBS without TPPU.

## Data Availability

Data are available from the corresponding author upon reasonable request.

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
