# Peer review of "Inhibition of Soluble Epoxide Hydrolase Does Not Promote or Aggravate Pulmonary Hypertension in Rats"

_cells, 2023, doi:10.3390/cells12040665_

Round 1
Reviewer 1 Report
This manuscript focuses on the effect of an sEH inhibitor in a SuHx rat model of pulmonary hypertension. Different effects in different models of PH like chronic hypoxia exposure in mice or monocrotaline-induced PH in rats have previously been reported. Along these lines the current investigation does not conclusively solve the reason for the discrepant findings but adds data from another model of PH (SuHx in rats).
The manuscript is well written and the authors have large expertise in the field of PH.
The following points should be addressed in detail:
I wonder why differences of the number of experiments exist among groups and different parameters assessed. E.g. for Figure 1 A ´SuHx´ an n of 10 is displayed whereas the ´Control´ consists of n=7 and the ´Control+TPPU´ group of n=6.
As a further example the ´SuHx+TPPU´ group in Figure 1B comprises n=8, in Figure 2A n=11 but in Figure 2B only n=4 and in Figure 2C n=5. This is even more puzzling as you obviously have investigated n=5 in the SuHx+TPPU group in Figure 2C but n=6 in this group in Figure 3B although bioth are derived from histological sections. This accounts also for other Figures/groups. Please clarify how the design of the animal study was and if you initially used the same group size for all 4 groups investigated (which should be the case). Please explain the different ´n´ throughout and also why for different parameters within the same group different ´n´ are given. This should also be integrated in the Methods section.
Please include all relevant literature e.g:
Revermann et al., J Hypertens 2009 Feb;27(2):322-31.
Keserü et al., Cardiovasc Res 2010 Jan 1;85(1):232-40.
In Figure 5E description of ´E´ is missing in the legend.
Author Response
We thank the reviewer for helpful comments
Point 1
“I wonder why differences of the number of experiments exist among groups and different parameters assessed. E.g. for Figure 1 A ´SuHx´ an n of 10 is displayed whereas the ´Control´ consists of n=7 and the ´Control+TPPU´ group of n=6.
As a further example the ´SuHx+TPPU´ group in Figure 1B comprises n=8, in Figure 2A n=11 but in Figure 2B only n=4 and in Figure 2C n=5. This is even more puzzling as you obviously have investigated n=5 in the SuHx+TPPU group in Figure 2C but n=6 in this group in Figure 3B although both are derived from histological sections. This accounts also for other Figures/groups. Please clarify how the design of the animal study was and if you initially used the same group size for all 4 groups investigated (which should be the case). Please explain the different ´n´ throughout and also why for different parameters within the same group different ´n´ are given. This should also be integrated in the Methods section.”
As underlined by the reviewer, the number of animals is slightly different between experiments. For Figures 1 and 2 in the SuHx+TPPU group, one rat died before and 2 rats died during hemodynamical measurements explaining the differences. AT/ET was only obtained in a subset of rats from each group as was also done for histological analyses. This is now specified in the legends of these figures. In addition, we analyzed the missing animals for pulmonary muscularization and the Figure 2C has been modified accordingly. Finally, data on Figures 4 and 5 were obtained from different series of animals than those presented in Figures 1 to 3. This is now specified in the legend of these Figures.
Point 2
“Please include all relevant literature e.g:
Revermann et al., J Hypertens 2009 Feb;27(2):322-31.
Keserü et al., Cardiovasc Res 2010 Jan 1;85(1):232-40.”
According with the reviewer, the previously published papers by Revermann et al and Keserü et al (references 9 and 17 respectively) are important works in the field of the impact of sEH inhibition in pulmonary hypertension and have been cited and discussed through the text of our manuscript (lines 54-58; 264-266;283-286)
Point 3
“In Figure 5E description of ´E´ is missing in the legend.”
D was in fact mentioned two times. This error has been corrected
Reviewer 2 Report
Dear authors,
this is an interesting work addressing the impact of soluble epoxide hydrolase (sEH) inhibition on experimental pulmonary hypertension (PH) in the well-established SU5416 / chronic hypoxia rat model of PH. The treatment was initiated after stable disease induction and showed no significant effects of the sEH inhibitor TPPU either on hemodynamics or vascular remodeling.
Although the results did not show beneficial effects in this model, the present study is interesting and nicely performed.
I have some minor comments that need to be clarified:
Comments:
line 78: You refer for the dosing of TPPU to citation 12. Please provide more information on bioavailability and plasma level here.
Figure 5: Interestingly, TPPU changes plasma level of the sEH metabolites. Please comment on the stronger increase within the SU/Hx model as compared to healthy controls. Does this suggest an upregulation of sEH in the SuHx model?
Author Response
We thank the reviewer for helpful comments
Point 1
“line 78: You refer for the dosing of TPPU to citation 12. Please provide more information on bioavailability and plasma level here.”
We did not quantify TPPU in the present work since a similar dose of 5 mg/L of drinking water in adult male Sprague-Dawley rats also used in this work was already shown to reach plasma concentrations much far higher than required for inhibiting sEH with a significant drug-target engagement demonstrated at the cardiac level. This aspect is now better described in the methods (lines 81-83).
Point 2
“Figure 5: Interestingly, TPPU changes plasma level of the sEH metabolites. Please comment on the stronger increase within the SU/Hx model as compared to healthy controls. Does this suggest an upregulation of sEH in the SuHx model?“
As stressed by the reviewer, the increase in plasma diol-to-epoxide ratio could have been related to an increased sEH expression. We did not measure sEH expression in the liver that mainly drives plasma levels of sEH substrates and metabolites, but we now specified in the discussion section that, at the opposite, a decrease in sEH expression has been previously shown to be induced by chronic hypoxia and did not probably contribute to the results obtained regarding plasma diol-to-epoxide ratio (lines 293-295).
Reviewer 3 Report
In this manuscript Leuiller and co-workers examine the soluble epoxide hydrolase inhibitor TPPU regarding its potential effect on pulmonary artery structure and function. They show that TPPU has no deleterious effects on pulmonary circulation, as it does not promote or aggravate pulmonary hypertension. The study includes both in vivo (animal) and in vitro (human tissue) part, presenting novel and original findings worth of publication. The manuscript is well-written, however minor alterations and corrections are needed.
In figure 4, SuHx-TPPU rats seem to present higher RV end-diastolic volume and lower ejection fraction compared to SuHx rats, but as far as I understand, only comparison versus normal controls has been performed (or is depicted), which yields a statistically significant difference. Comparison between SuHx-TPPU and SuHx values for RVEDD and RVEF should be performed and clearly presented, including p-values, either in the text (lines 198-205) or in the figure/figure legend. Do the authors believe that a direct effect of TPPU on myocardial contractility is possible? A comment on potential effects of sEH inhibition on cardiac muscle cells should be included in the discussion
.Minor corrections:
abstract lines 23-25: expression needs to be improved/simplified, for example "Treatment with TPPU started 5 weeks after SU5416 injection. No differences regarding the increase in pulmonary vascular resistance, remodeling, and inflammation nor the abolishment of phenylephrine-induced pulmonary artery constriction were noted in SuHx rats.
line 29: does instead of did
line 53: evidence insead of evidences
lines 54-55: ... on the acute inflammation process known to contribute to the development of PH... instead of ...on the acute inflammation that are known to contribute the development of PH...
lines 73-74: ...for 5 additional weeks... instead of ...for additional 5 weeks...
line 106 and 286: abbreviations should be used consistently (RV)
line 112: 10-5 (-5 should be superscript)
line 258: evident instead of evidenced
Author Response
We thank the reviewer for helpful comments
Point 1
“In figure 4, SuHx-TPPU rats seem to present higher RV end-diastolic volume and lower ejection fraction compared to SuHx rats, but as far as I understand, only comparison versus normal controls has been performed (or is depicted), which yields a statistically significant difference. Comparison between SuHx-TPPU and SuHx values for RVEDD and RVEF should be performed and clearly presented, including p-values, either in the text (lines 198-205) or in the figure/figure legend.
As suggested by the reviewer, although no significant effects of TPPU was detected, we now provided the P-values in all figures.
Point 2
Do the authors believe that a direct effect of TPPU on myocardial contractility is possible? A comment on potential effects of sEH inhibition on cardiac muscle cells should be included in the discussion.
As requested by the reviewer, we added in the discussion a short paragraph describing the protective effects of sEH inhibition on cardiac contractility, in particular during ischemia-reperfusion injury, through EET-mediated modulation of L-type calcium and ATP-sensitive potassium channels (lines 307-310) and added the corresponding reference [22].
Point 3
Minor corrections:
abstract lines 23-25: expression needs to be improved/simplified, for example "Treatment with TPPU started 5 weeks after SU5416 injection. No differences regarding the increase in pulmonary vascular resistance, remodeling, and inflammation nor the abolishment of phenylephrine-induced pulmonary artery constriction were noted in SuHx rats.
line 29: does instead of did
line 53: evidence instead of evidences
lines 54-55: ... on the acute inflammation process known to contribute to the development of PH... instead of ...on the acute inflammation that are known to contribute the development of PH...
lines 73-74: ...for 5 additional weeks... instead of ...for additional 5 weeks...
line 106 and 286: abbreviations should be used consistently (RV)
line 112: 10-5 (-5 should be superscript)
line 258: evident instead of evidenced
All these points have been corrected in the manuscript.
Round 2
Reviewer 1 Report
The author shave addressed all of my concerns
Author Response
None